# Deformation and Simulation of the Cellular Structure of Foamed Polypropylene Composites

**DOI:** 10.3390/polym14235103

**Published:** 2022-11-24

**Authors:** Wei Gong, Di Zhang, Chun Zhang, Xiangbu Zeng, Li He, Tuanhui Jiang

**Affiliations:** 1School of Materials and Architectural Engineering, Guizhou Normal University, Guiyang 550025, China; 2National Engineering Research Center for Composite Modified Polymer Materials, Guiyang 550014, China; 3School of Mathematical Sciences, Guizhou Normal University, Guiyang 550025, China

**Keywords:** foamed polypropylene, cell deformation, software simulation, temperature

## Abstract

Foamed Polymer is an important polymer material, which is one of the most widely used polymer materials and plays a very important role in the polymer industry. In this work, foamed polypropylene (PP) composites are prepared by injection molding, and the cell deformation process within them is studied by combining visualization technology and COMSOL software simulation. The results shows that the deformation of isolated cells depends in temperature, and there is no macroscopic deformation. There was no significant difference between the stress around adjacent cells at different temperatures, but the stress at different positions around the adjacent cells has obvious changes, and the maximum stress at the center of the adjacent cells was 224.18 N·m^−2^, which was easy to cause a lateral deformation of the cells. With the increase in temperature, the displacement around the adjacent cell gradually increased, the maximum displacement of the upper and lower symmetrical points of the cell was 14.62 μm, which is most likely to cause longitudinal deformation of the cell; the deviation of the cell deformation parameter gradually increased, which led to deformation during the growth of the cell easily. The simulation results were consistent with the visualized cell deformation behaviors of the foamed PP composites.

## 1. Introduction

Polymer microcellular-foamed plastics are polymer foam materials with cell diameter between 0.1 and 10 μm, and cell densities greater than 10^8^ cells∙cm^−3^, The uniform and fine cell structure makes the polymer microcellular foamed materials stand out from non-foamed materials with the advantage of comprehensive performances, which played an important role in many application fields [1,2,3,4]. The comprehensive performances of the foamed material, such as mechanical properties and thermal insulation properties, and also electrical conductivity and electromagnetic shielding properties [5,6], are mainly affected by the cellular structure, while the cell deformation during the foaming process is one of the most important factors in determining the final cell structure. Therefore, improving the foaming quality by controlling and optimizing the cell deformation of the foaming materials is conducive to improving the overall performance, which not only promotes the development of high-quality lightweight materials, but also provides a theoretical basis for future research on improving the quality of foamed materials.

In recent years, researchers have carried out extensive research on the cell deformation and simulation in polymer foams. Tromm et al. [7] investigated the cell structure changes in polystyrene/carbon dioxide systems during high-pressure foam injection molding by in situ visualization techniques. They found that the filling pressure and forming time were the main parameters of the size and shape on the nucleated cell in the melt, the higher filling pressure and the longer filling time, and the smaller the cell size the larger cell deformation. Ataei et al. [8] studied the influence of viscosity distribution on the deformation and growth of cells with the Lattice Boltzmann model. They found that viscosity was the key factor affecting cell deformation and micro-cell formation, and the increase in the viscosity under different capillary number reduces the cell deformation. Wong et al. [9] successfully assembled a visualization system with precise program-controlled heating/cooling system, which has been used to study the influence of polymer crystal formation on cell formation and growth. It was found that with decreasing temperature, the crystals gradually formed and grew into spherulites. This system can be effectively applied to study the relationship between crystallization kinetics and cell nucleation, growth, and deterioration during the foaming process. Mirzaee et al. [10] predicted the cell morphology and the heterogeneous deformation of cell boundaries based on the “cellular model” and the “conservation of gas content” in the polymer, by combining Influenced Volume (IVA) and Finite Element 2D models. The results show that the simulation results are basically consistent with the final cell form of the extrusion foaming experiment. Wang [11] observed the rotation and deformation of fibers with different characteristics around the cells during free foaming and injection foaming and established mathematical models of the cell affecting area and cell deformation during cell growth process.

Zhu et al. [12] studied the coarsening of two adjacent cells with different sizes during polymer foaming by finite element simulations. Based on quadratic triangle finite element analysis and implicit time evolution scheme, the governing diffusion equation was solved in axisymmetric coordinate system, and the effects of gas concentration, cell spacing and initial cell size on cell coarsening were simulated. Higher gas concentrations were found to be beneficial in preventing cell coarsening, and shorter distances and larger sizes between adjacent cells would promote cell coarsening. Zhang [13] used COMSOL finite element simulation software to construct a two-dimensional heat transfer model of polymer foamed materials and calculated the relationship between the thermal insulation performance of foamed polymer materials and the distribution of cells. The results show that the uniform cell structure makes the thermal insulation performance of the material better than that of the random cell structure. Upreti et al. [14] studied the deformation of aluminum 5052 hexagonal honeycomb core using the finite element software Ansys. The results show that the deformation and equivalent stress decreased with the increase in plate thickness. Niedziela, D., et al. [15] studied heterogeneity in the distribution of foam fraction in chemically expanding blown polyurethane foam. The nonlinear coupled system of partial differential equations governing flow was numerically solved using finite-volume techniques, and the associated results are presented and discussed with graphical illustrations. The models were validated with experimental data, and simulation results favorably compared with the experiment observations. The above research focused on the physical properties of the injection molding process, cell size and foaming environment through visualization technology, software and mathematical models. The relevant research results provide guidance for in-depth understanding of cell deformation during polymer foaming. However, most of the above research is based on the final cell structure to deduce the cell deformation. There are no reports combining visualization techniques and software simulations to directly observe and study the effect of temperature on cell deformation. 

In this study, the effect of temperature on the cell deformation of the foamed polypropylene (PP) is observed directly by using visualization technology, and the cell deformation process is simulated by eliminating the deformation error of independent cells and adjacent cells. It can control the cell deformation by temperature, the research flow chart is shown in Figure 1. The relevant results provide a theoretical and experimental basis for the improvement of foaming quality, which is expected to facilitate the industrial application of lightweight polymer products. 

## 2. Experiment

### 2.1. Experimental Materials

PP (L5E89) with a melt flow rate (MFR, 2.16 kg/190 °C) of 3.1 g·10 min^−1^ was provided by China National Petroleum Corporation (Beijing, China). The foaming agent masterbatch was self-prepared, and the preparation was described in our previous work [16]. An injection molding machine equipped with visualization system (TTI-205Ge, Dongguan Donghua Machinery Co. Ltd., Dongguan, China) was used to observe the cell growth and deformation in foaming (Figure 2a). A self-designed visual mold [17] was utilized, in which two transparent sapphire sheets were mounted in the center of the mold fixed plate and the moving plate, and a high-speed microscope camera (TK-C1031EC, Japan JVC company, Yokohama, Japan) and a light source were mounted on the mold fixed plate and the moving plate, respectively (Figure 2b).

### 2.2. Visual Injection Foaming

The PP and the foaming agent masterbatch were uniformly blended by mass fraction of 100:6. The foamed PP samples were prepared by pressure release molding at 185 °C, 195 °C and 205 °C, respectively. The foaming processing parameters are shown in Table 1.

### 2.3. Testing and Characterization

#### 2.3.1. Characterization of the Cell Deformation Parameters

The selection criteria for foam samples are the video screenshots of the cell growing process at each temperature, the cells in the screenshots are the research objects. Cell deformation parameters(D) [18,19,20] are given by Equation (1).
(1)$$D=\frac{A-B}{A+B}$$

Visual video is captured using video processing software at 25FPS. The axial lengths A and B of the cells were measured by picture analysis software. Figure 3a is the schematic diagram of an isolated cell, and the schematic diagram of the adjacent cells is shown in Figure 3b.

#### 2.3.2. Rheological Test

A rotational rheometer (HAAKE Mars60, Thermo Fisher Scientific, Waltham, MA, USA) was used to test the dynamic rheological properties of the samples. The thickness of the samples was 1 mm, the diameter was 20 mm, the strain was set to 1%, the cooling rate was 0.1667 °C∙S^−1^, and the sweep frequency was 0.1 Hz [21]. 

## 3. Influence of Injection Temperature on the Cell Deformation of Foamed PP Composites and the Corresponding Simulation

### 3.1. Influence of Injection Temperature on an Isolated Cell Deformation of Foamed PP Composites

Because PP is a semi-crystalline polymer, an isolated cell deformation of the foamed PP materials obtained at a processing temperature of 185 °C to 205 °C is studied according to the results of the previous experiments.

#### 3.1.1. An Isolated Cell Deformation at an Injection Temperature of 185 °C

Figure 4 shows the digital images of the deformation process of an isolated cell at an injection temperature of 185 °C. It can be seen from the figure that with the increase in molding time, the cell gradually grows, and the shape of the isolated cell appears to be round throughout the growth process. In addition, almost no macroscopic deformation of the sample is observed during the whole growth process, and the cell deformation parameter (D) of the isolated cell is basically maintained within 0.017 (Figure 5a). During the free growth of the cell, the cell deformation parameter increases with the molding time, and there is no stable growth trend of cell deformation over time as it varies randomly (Figure 5b).

#### 3.1.2. An Isolated Cell Deformation at an Injection Temperature of 195 °C

Figure 6 and Figure 7 exhibits the digital images of the deformation process of an isolated cell at an injection temperature of 195 °C, the structural parameters (D) of a few cells (D) ≧ 0.017, as shown in the red circle (Figure 7a). The trends of cell growth and deformation at 195 °C are similar to those at 185 °C.

#### 3.1.3. An Isolated Cell Deformation at an Injection Temperature of 205 °C

Figure 8 shows the digital images of the deformation process of an isolated cell at an injection temperature of 205 °C. The isolated cell grows almost centro-symmetrically throughout the foaming process, and again no local macroscopic deformation of the samples is observed. Figure 9 shows the cell deformation parameter of the isolated cell at different molding times. It can be seen from Figure 9a that the cell deformation parameter of the isolated cell is basically maintained within 0.018, and the increase in temperature has little effect on the cell deformation parameter of the isolated cell. The trends of cell growth and deformation at 205 °C were similar to those at 185 °C and 195 °C. 

The effect of the molding temperature on the cell deformation parameter and cell deformation stability at different molding times are summarized, as shown in Table 2. The cell deformation is unstable at 185 °C, 195 °C and 205 °C; with the increase in molding time it presents random changes, and there is no local macroscopic deformation; the cell deformation parameter is small at 205 °C, and the cell deformation trend is not obvious.

### 3.2. Influence of Injection Temperature on Adjacent Cells Deformation of Foamed PP Composites

#### Adjacent Cells Deformation at Different Injection Temperature

Through the above analysis of an isolated cell deformation it is found that the cell deformation parameter (D) of 0.018 is the reference value for studying the deformation of adjacent cells, and the following research and analysis of adjacent cells will be based on this reference value.

Figure 10 is the digital screenshot of the adjacent cells’ deformation with the same size at 185 °C. It is found from the figure that the adjacent cells’ deformation becomes more and more serious as the cells grow. With the increase in molding time, the cell deformation parameters (D) gradually increase, and the cell deformation parameters of adjacent cells are basically the same due to the same pressure inside the cells (as shown in Figure 11). The trends of cell growth and deformation at 195 °C and at 205 °C are similar to those at 185 °C (as shown in Figure 12, Figure 13, Figure 14 and Figure 15).

In conclusion, with the increase in molding temperature the deformation of adjacent cells with the same size becomes more and more serious. At 185 °C the macroscopic deformation occurs after 6.84 s at 195 °C the macroscopic deformation occurs after 5.13 s; at 205 °C the macroscopic deformation occurs after 3.42 s; the forming time of macroscopic deformation occurs earlier. At any temperature, with the increase in molding time, the cell deformation parameters gradually increase, which indicates that the cells’ deformation was gradually serious. The main reason is that with the increase in injection temperature the displacement of resin on the cell wall gradually increases, so the cell deformation becomes more and more serious (see the analysis section of simulation results).

### 3.3. Finite Element Modeling and Analysis of the Experimental Results

#### 3.3.1. Finite Element Modeling

(1)Establishment of solid model
Using the standard linear solid model, its creep model is shown in Figure 16a [21].
(2)Boundary conditions

A two-dimensional model of cell growth was established by setting fixed constraints on the boundary of resin. The same pressure load is applied inside the cell to simulate the influence of different temperatures on the cell deformation, as shown in Figure 16b.
(3)Determination of physical parameters of the foamed PP composites

In the actual injection molding process, it is found that the deformation rules of the cells are different at 185 °C, 195 °C and 205 °C, considering that the viscoelasticity of PP resin would change at different temperatures, which might affect the deformation of the cells. Therefore, the viscosity and shear modulus of the PP from140 °C to 240 °C are measured by a rotational rheometer at a constant frequency, as shown in Figure 17.

According to the relationships between temperature and shear modulus, as well as temperature and viscosity in Figure 17, the detailed data of shear modulus and viscosity at 185 °C, 195 °C and 205 °C are shown in Table 3.

#### 3.3.2. Finite Element Simulation of the PP Viscoelasticity on the Cell Deformation at Different Temperatures

Within a space of 600 μm radius in the resin, a two-dimensional model of two adjacent cells with an initial radius 100 μm is established. The center distance of the two cells is 300 μm, and the same pressure, 100 Pa, is applied inside the cells. Using the finite element software COMSOL the physical parameters of Table 3 are used as inputs; the time is set to 1 s, and the effect of PP viscoelasticity on the cell deformation and the stress and displacement changes around the cells are simulated at different temperatures.

It can be seen from Figure 18 and Figure 19 that the stress in the system hardly changes with the change in injection temperature. The stresses are basically distributed in the range of 129.15–224.18 N·m^−2^ at different temperatures.

It can be seen from Figure 20 and Figure 21 that the injection temperature has a significant effect on the displacement of the adjacent cells; the displacement of the resin in the cell wall gradually increases with the decrease in viscosity caused by the increase in temperature. The displacements of the resin in the cell wall are in the range of 3.08–8.82 μm at 185 °C, 3.89–11.16 μm at 195 °C, and 5.09–14.62 μm at 205 °C, respectively. Under the same pressure in the cells, the higher the resin viscosity, the greater the resin displacement of the cell wall. Therefore, the displacement range of the resin is larger at 205 °C, and the cells are more prone to deformation.

In order to more clearly reflect the influence of the temperature on the displacement, stress and deformation of the adjacent cells, we build the different regions model around adjacent cells. The circumference (Length) of the model is 628 μm, as shown in Figure 22. The simulation results are shown in Figure 23.

It can be seen from Figure 23a that there is no significant difference in the change in stress around the adjacent cells at different temperatures, and only slight changes can be observed when the stress curves are zoomed in. The stress between the adjacent cells gradually increases with temperature. However, there is a significant difference in the pressure at different positions around the cell. The maximum stress at the point c (the center of the adjacent cells) is 224.18 N·m^−2^; the minimum stress at the points b and d (the upper and lower symmetrical points of the cells) is 129.15 N·m^−2^; the stress at the point a is 173.33 N·m^−2^. The above results indicate that lateral deformation of cells is most likely to occur at the center of the adjacent cells. However, the temperature has a significant effect on the displacement around adjacent cells, as shown in Figure 23b. There is a significant difference in displacement at different positions around the cells at different temperatures, and the displacement around adjacent cells gradually increases with temperature. 

Taking the case at 205 °C as an example, the minimum displacement at the point c is 5.09 μm; the maximum displacement at the points b and d is 14.62 μm; the displacement at the point a is 11.86 μm, which indicates that longitudinal deformation of the cells is most likely to occur in the adjacent cells in the upper and lower symmetric regions.

The cell deformation parameters at different temperatures are obtained by simulating the deformation of the adjacent cells. It can be seen from Table 4 that the cell deformation parameter increases gradually with the increase in temperature. Considering the single factor of resin viscoelasticity, the increase in temperature leads easily to cell deformation in the process of the cell growth. When the injection temperature is 195 °C, the minimum cell deformation parameter is 0.018, and the cell deformation is relatively difficult. The simulation results are consistent with the cell deformation behaviors at the tested temperatures.

## 4. Conclusions

In this work, the visualization technology and COMSOL software were used to study the deformation process of the cell. The results showed the deformation of isolated cells without macroscopic deformation. The stress at different positions around the adjacent cells has obvious changes, which can easily cause the transverse deformation of the cell. With the increase in temperature, the displacement around the adjacent cells gradually increases the maximum displacement of the upper and lower symmetrical points of the cell, which is most likely to cause longitudinal deformation of the cell. The software simulation results are basically consistent with the visualized cell deformation behavior of PP foam composites.

## Figures and Tables

**Figure 1 polymers-14-05103-f001:**
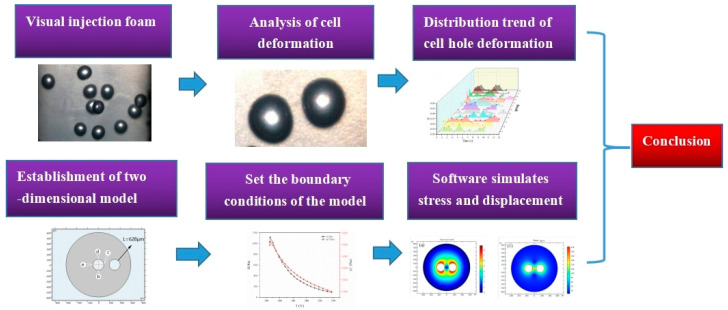
Flowchart of the research process.

**Figure 2 polymers-14-05103-f002:**
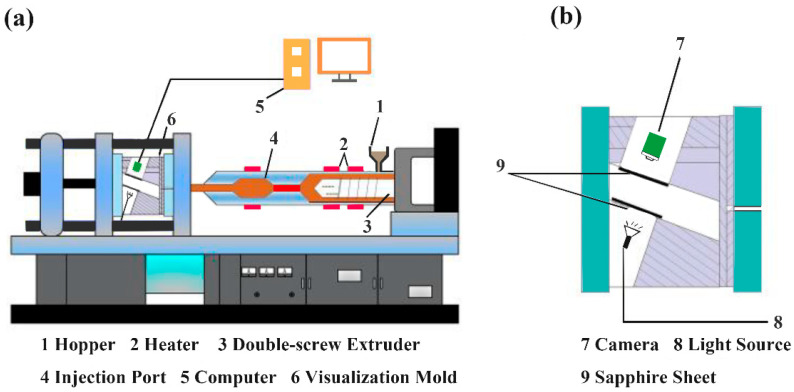
Schematic diagrams of the: (**a**) injection molding machine, and (**b**) visual injection mold.

**Figure 3 polymers-14-05103-f003:**
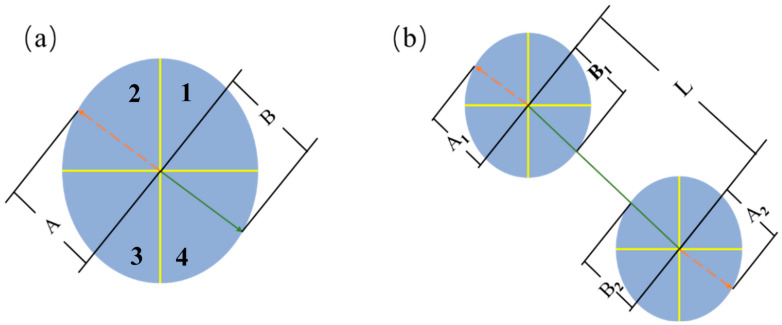
Schematic diagrams of the axial lengths of a single cell (**a**) and two adjacent cells (**b**).

**Figure 4 polymers-14-05103-f004:**
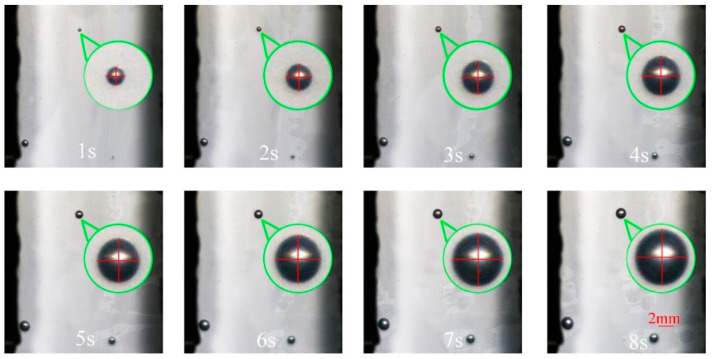
Digital images of the deformation process of an isolated cell at 185 °C.

**Figure 5 polymers-14-05103-f005:**
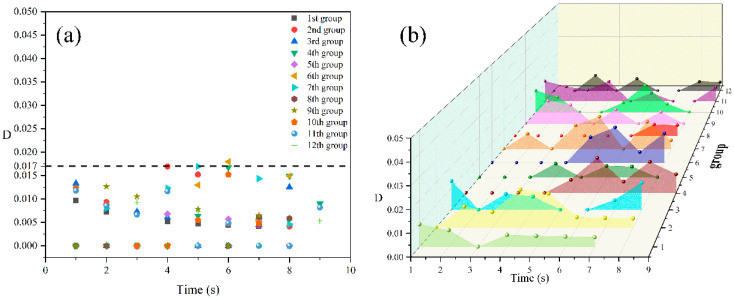
Distribution (**a**) and trend (**b**) of the cell deformation parameters of an isolated cell at 185 °C.

**Figure 6 polymers-14-05103-f006:**
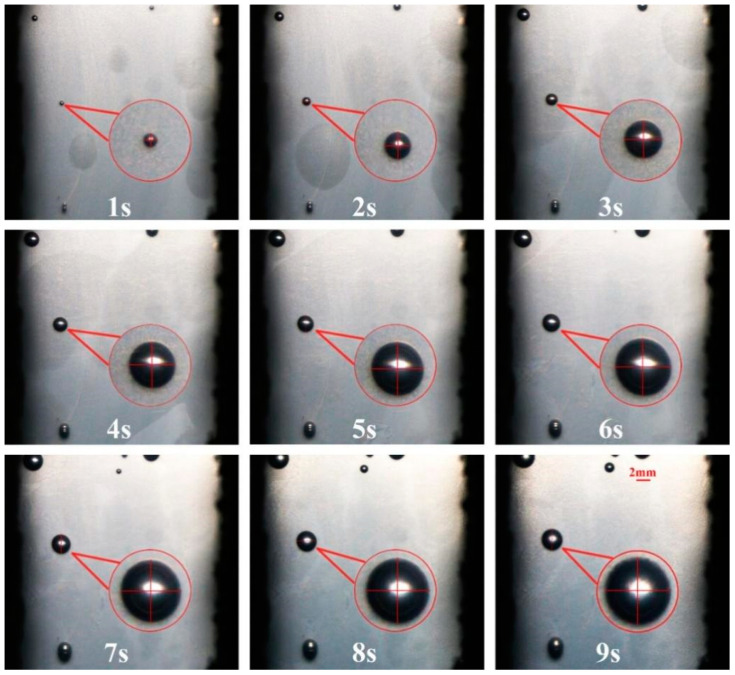
Digital images of the deformation process of an isolated cell at an injection temperature of 195 °C.

**Figure 7 polymers-14-05103-f007:**
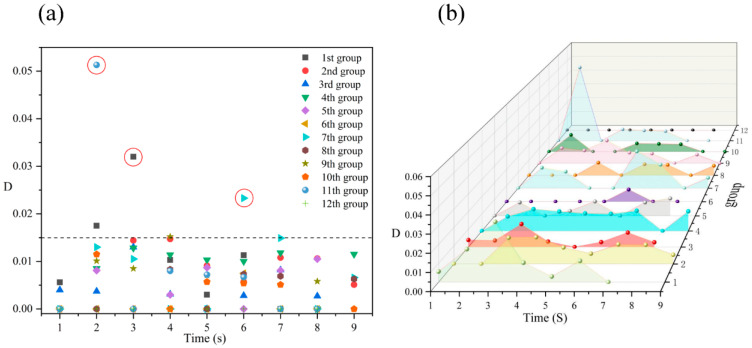
Distribution (**a**) and trend (**b**) of the cell deformation parameters of an isolated cell at 195 °C.

**Figure 8 polymers-14-05103-f008:**
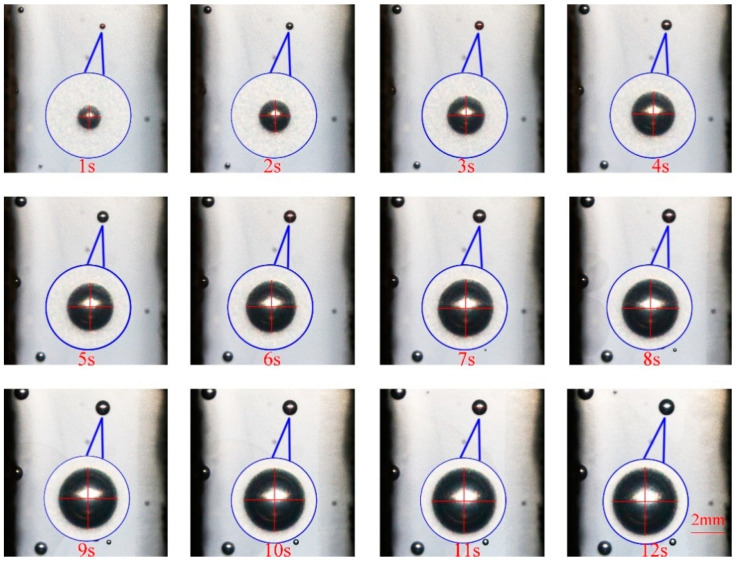
Digital images of the deformation process of an isolated cell at an injection temperature of 205 °C.

**Figure 9 polymers-14-05103-f009:**
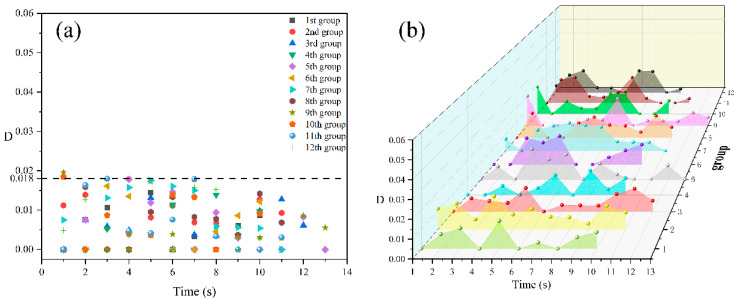
Distribution (**a**) and trend (**b**) of the cell deformation parameters of an isolated cell at 205 °C.

**Figure 10 polymers-14-05103-f010:**
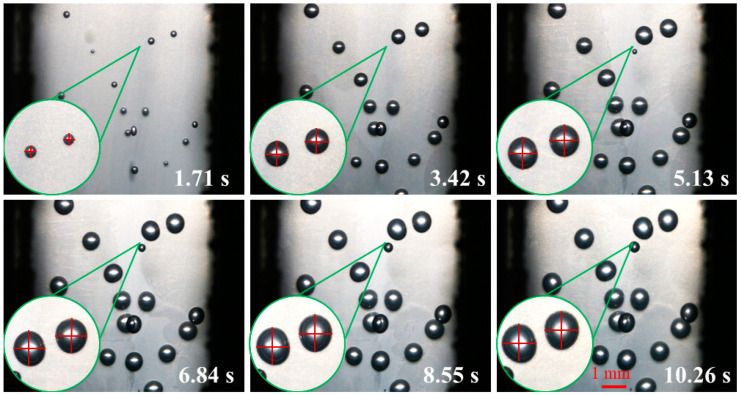
Digital images of the deformation process of adjacent cells with the same size at 185 °C.

**Figure 11 polymers-14-05103-f011:**
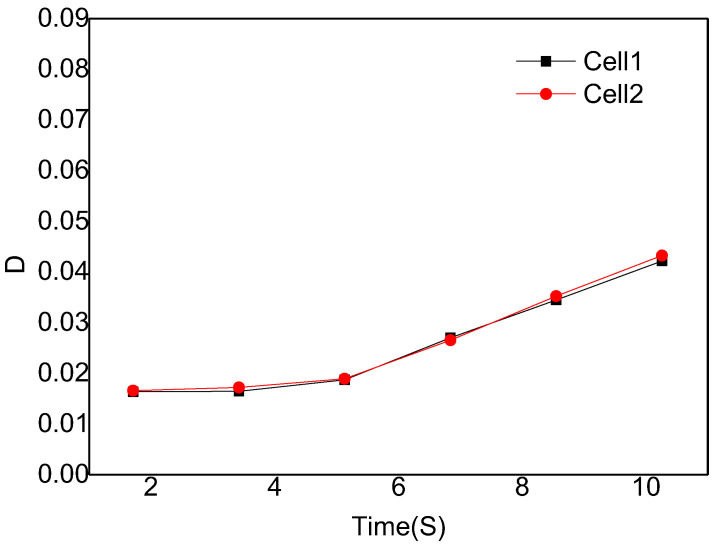
Deformation influence of adjacent cells with the same size at 185 °C.

**Figure 12 polymers-14-05103-f012:**
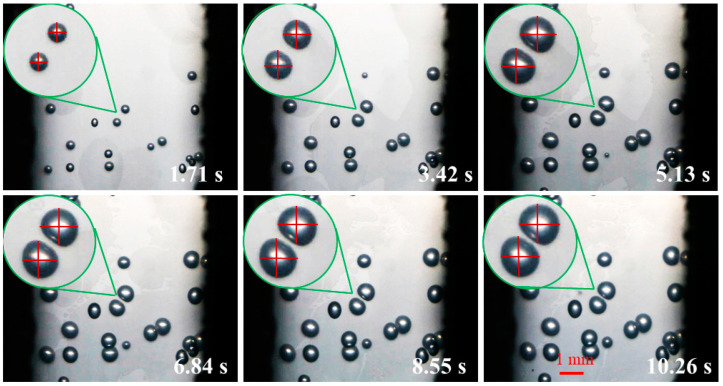
Digital images of the deformation process of adjacent cells with the same size at 195 °C.

**Figure 13 polymers-14-05103-f013:**
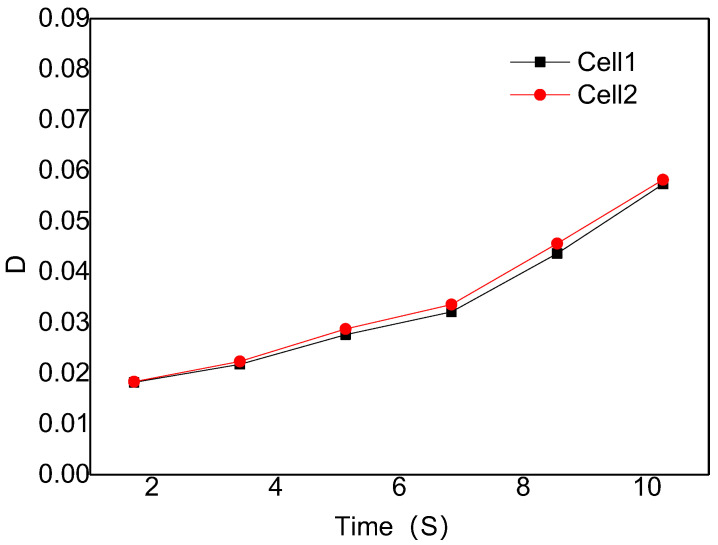
Deformation influence of adjacent cells with the same size at 195 °C.

**Figure 14 polymers-14-05103-f014:**
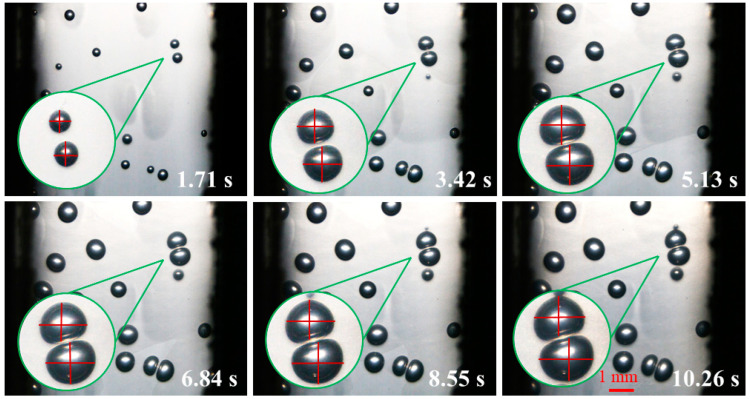
Digital images of the deformation process of adjacent cells with the same size at 205 °C.

**Figure 15 polymers-14-05103-f015:**
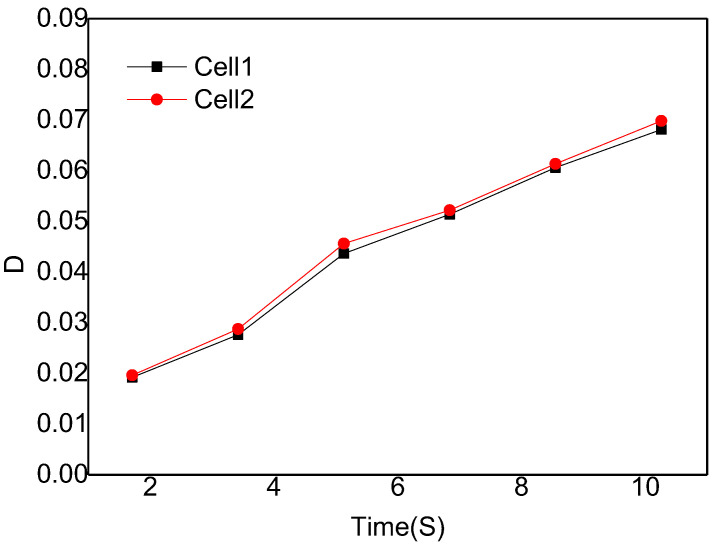
Deformation influence of adjacent cells with the same size at 205 °C.

**Figure 16 polymers-14-05103-f016:**
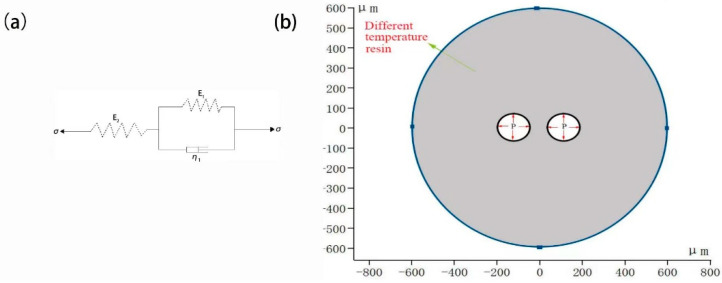
(**a**) Standard linear solid model and (**b**) model boundary conditions.

**Figure 17 polymers-14-05103-f017:**
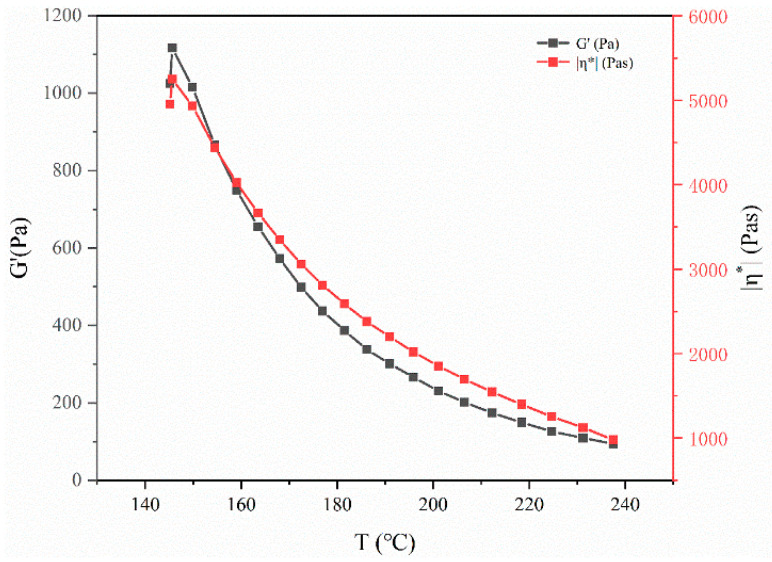
Shear modulus and viscosity of polypropylene materials at different temperatures.

**Figure 18 polymers-14-05103-f018:**
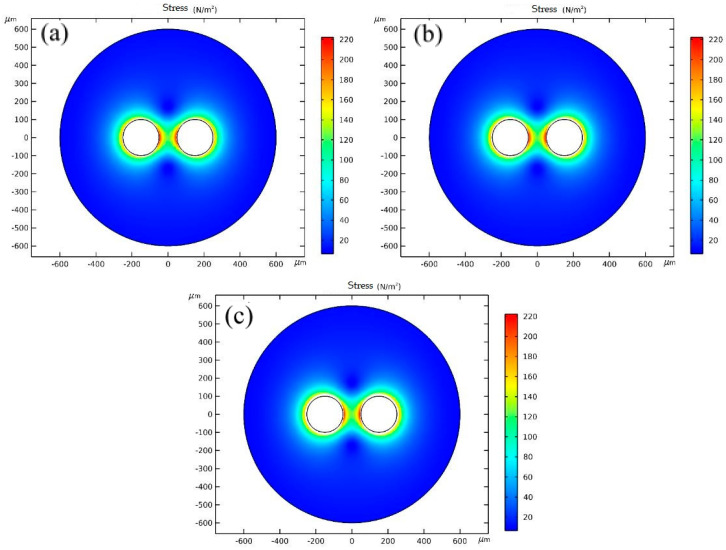
Stress cloud diagrams of any two adjacent cells at different temperatures: (**a**) 185 °C; (**b**) 195 °C, (**c**) 205 °C.

**Figure 19 polymers-14-05103-f019:**
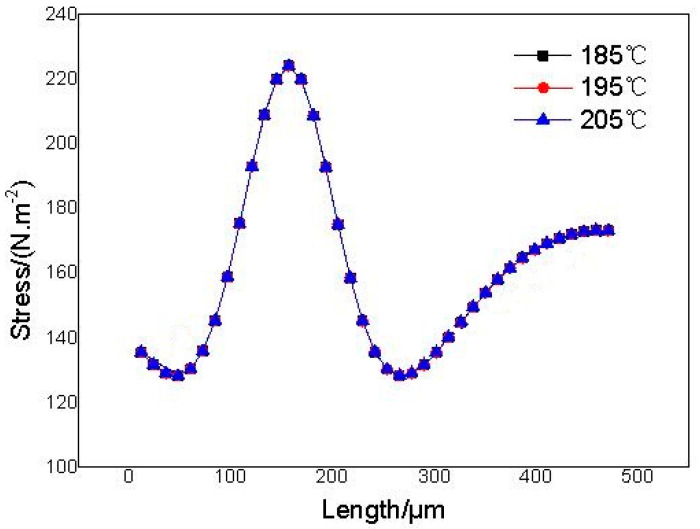
Stresses between any two adjacent cells at different temperatures.

**Figure 20 polymers-14-05103-f020:**
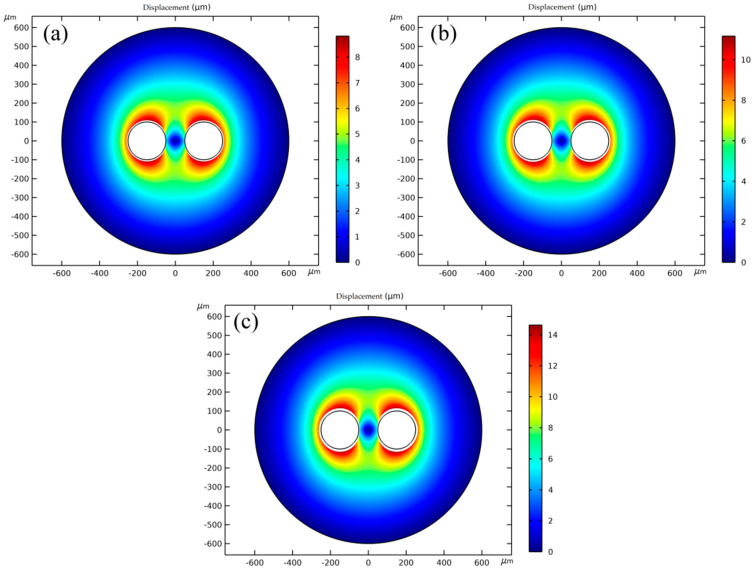
Displacement cloud diagrams of any two adjacent cells at different temperatures: (**a**) 185 °C; (**b**) 195 °C; (**c**) 205 °C.

**Figure 21 polymers-14-05103-f021:**
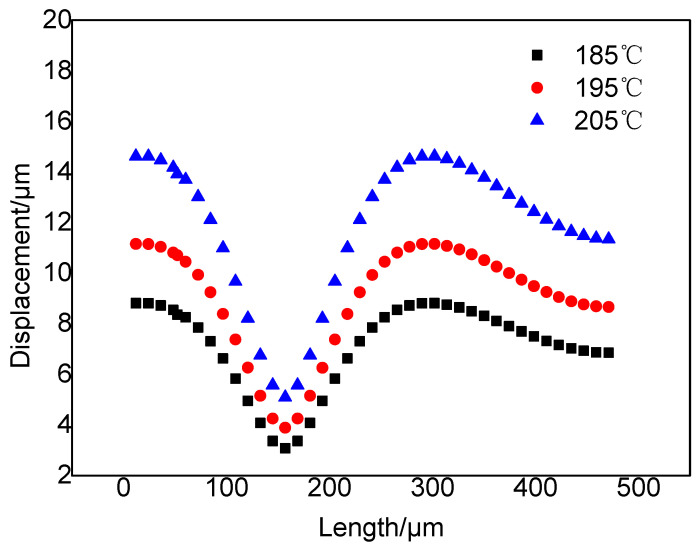
Displacements of any two adjacent cells at different temperatures.

**Figure 22 polymers-14-05103-f022:**
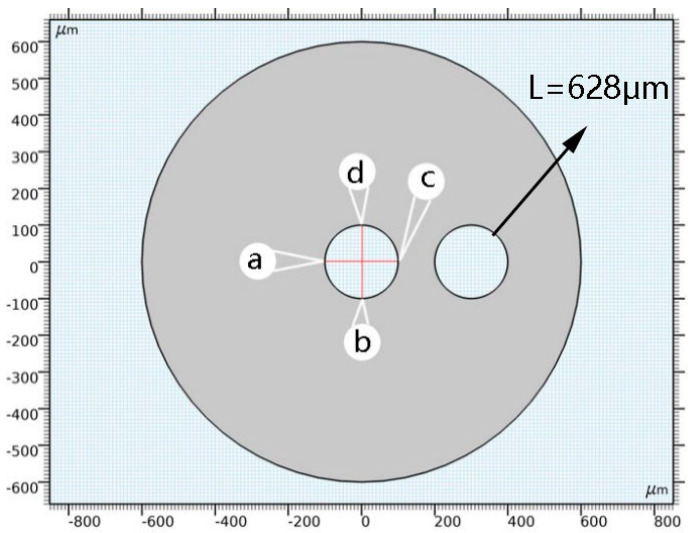
Model diagram of any point around the cell.

**Figure 23 polymers-14-05103-f023:**
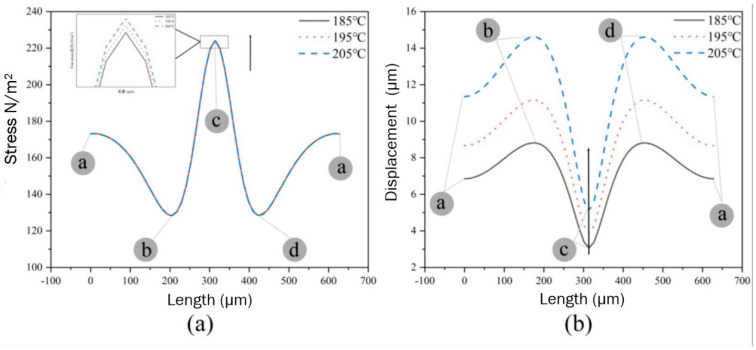
Stress (**a**) and displacement (**b**) of any point around the adjacent cells at different temperatures.

**Table 1 polymers-14-05103-t001:** Processing parameters of the foamed PP samples.

**Foaming Processing Parameters**	**Injection Speed** **(mm/s)**	**Injection** **Pressure** **(bar)**	**Injection** **Volume** **(mm)**	**Mold Opening Distance** **(mm)**	**Mold Temperature** **(°C)**	**Cooling Time** **(s)**
	30	40	22	0.6	40	30

**Table 2 polymers-14-05103-t002:** Cell deformation parameter and cell deformation stability at different temperatures.

No.	T (°C)	Cell Deformation Parameter (D)	Cell Deformation Stability
1	185	≤0.017	random variation
2	195	≤0.015	random variation
3	205	≤0.018	random variation

**Table 3 polymers-14-05103-t003:** Rheological parameters of the foamed PP materials.

Rheological Parameters	Density/g·m^−3^	Poisson’s Ratio	Young’s Modulus/Pa	Temperature/°C	Shear Modulus/Pa	Viscosity/Pa·s
Foamed PP	0.91	0.34	713	185	338	2380
				195	266	2020
				205	202	1697

**Table 4 polymers-14-05103-t004:** Cell deformation parameters of the spherical cell at different temperatures.

No.	T (°C)	A (μm)	B (μm)	D
1	185	108.67	103.89	0.022
2	195	106.85	103.08	0.018
3	205	111.35	105.09	0.029

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
