# Peer review of "Deformation and Simulation of the Cellular Structure of Foamed Polypropylene Composites"

_polymers, 2022, doi:10.3390/polym14235103_

Round 1
Reviewer 1 Report
The aim of this paper is to analyze the influence of temperature on the deformation of foamed polypropylene composites using visualization techniques and the numerical software COMSOL.
The analysis of the parameters that influence the deformation of the structure of polymer microcellular foamed materials is nowadays an interesting subject from a scientific and industrial point of view, however, unfortunately, the research developed by the authors presents certain limitations.
1.- The authors propose a method to measure the evolution of bubbles in foamed PP polymer, based on the use of digital images and simulation systems. Additionally, the authors evaluate the deviation of isolated cells belonging to various groups over time.
Unfortunately, foam injection is a manufacturing process with high variability and randomness. The authors do not indicate in the paper which is the selection criteria for the test specimen bubbles for each of the temperatures analyzed in the experimental tests. Without rigorous criteria for carrying out the experimental test, it is impossible to compare the bubble deformation results.
2.- The authors present a two-dimensional finite element model, based on the analysis of deformation between two adjacent cells with certain boundary conditions. Unfortunately, the authors do not detail why they selected these boundary conditions (dimensions and pressure inside the cells) to perform the analysis.
3.- The research presented compares the experimental and numerical results. Unfortunately, they don't present an analytical match at any of the temperatures tested.
4.- Authors are recommended to check the article's writing and spelling.
5- Authors are recommended not to include formulas in the background of the paper.
Author Response
We must thank you and the reviewer for the critical feedback on our manuscript entitled "Deformation and Simulation of the Cellular Structure of Foamed Polypropylene Composites" (Manuscript Number:polymers-1984982). We really appreciate all the valuable comments from the reviewers, which not only helped us to improve the quality of our manuscript, but also suggested some good ideas for future studies. Now we have added some experimental results and carefully revised the whole manuscript according to reviewers’ comments. The revised manuscript has been highlighted in blue in the revised manuscript. Below you will find our point-to-point responses to the reviewers’ comments and questions.
Response to Reviewer 1 Comments
Point 1: The authors propose a method to measure the evolution of bubbles in foamed PP polymer, based on the use of digital images and simulation systems. Additionally, the authors evaluate the deviation of isolated cells belonging to various groups over time.Unfortunately, foam injection is a manufacturing process with high variability and randomness. The authors do not indicate in the paper which is the selection criteria for the test specimen bubbles for each of the temperatures analyzed in the experimental tests. Without rigorous criteria for carrying out the experimental test, it is impossible to compare the bubble deformation results.
Response 1: Thanks for your good advice. The selection criteria for foam samples are the video screenshots of cell growing process at each temperature,the cell in the screenshots are the research objects.
Point 2: The authors present a two-dimensional finite element model, based on the analysis of deformation between two adjacent cells with certain boundary conditions. Unfortunately, the authors do not detail why they selected these boundary conditions (dimensions and pressure inside the cells) to perform the analysis.
Response 2: Thanks for your good suggestion.The diameter of the cell was chosen as 100μm, because the average diameter of the cell in the video screenshot was between 90μm and 110μm. Through the calculation of the ideal gas state equation, the gas phase pressure is 100Pa, so the pressure inside the cell is 100Pa during the simulation.
Point 3: The research presented compares the experimental and numerical results. Unfortunately, they don't present an analytical match at any of the temperatures tested.
Response 3: Thanks for your good advice. The matching experimental results and analysis have been supplemented in manuscript 3.2
Point 4: Authors are recommended to check the article's writing and spelling.
Response 4: Sorry for our poor expression. According to your suggestion, The writing and spelling of the article have been carefully revised and annotated in blue.
Point 5:Authors are recommended not to include formulas in the background of the paper.
Response 5:Thanks for your good suggestion.we have used language instead of formula in the background of the paper.

Reviewer 2 Report
In this work authors combine experiments and simulation to study the cell deformation in foamed polypropylene composites. The visualization of the cell deformations along with simulation results help to understand the influence of the temperature on the deformation of adjacent cells. Methodological aspects and results are clearly written and conclusions are back by the experiments and simulations. In my opinion, this type of work is suitable for publication in Polymers but a revision must be performed before publication. Thus, the text must be extensively revised to correct some English grammar (e.g., “they thus plays” (line 29) -> “they thus play”) and several typos (e.g., “10^8 cells/cm, The uniform” -> “10^8 cells/cm. The uniform” (line 27); or “deformation. the stress” -> “deformation. The stress” (line 260); or “which is easy to cause easy to cause” (line 262)). Also, in some figures (e.g., figures 11 and 13) Chinese symbols should be changed to Latin symbols. On the other hand, authors could explain the meaning of eq. 1 and justify why it is written there. Likewise, authors could mention why they use certain parameter values and not others: why the strain was set 1% and the frequency 0.1 Hz for the rheological test (section 2.3.2)?
Author Response
We must thank you and the reviewer for the critical feedback on our manuscript entitled "Deformation and Simulation of the Cellular Structure of Foamed Polypropylene Composites" (Manuscript Number:polymers-1984982). We really appreciate all the valuable comments from the reviewers, which not only helped us to improve the quality of our manuscript, but also suggested some good ideas for future studies. Now we have added some experimental results and carefully revised the whole manuscript according to reviewers’ comments. The revised manuscript has been highlighted in blue in the revised manuscript. Below you will find our point-to-point responses to the reviewers’ comments and questions.
Response to Reviewer 2 Comments
In this work authors combine experiments and simulation to study the cell deformation in foamed polypropylene composites. The visualization of the cell deformations along with simulation results help to understand the influence of the temperature on the deformation of adjacent cells. Methodological aspects and results are clearly written and conclusions are back by the experiments and simulations. In my opinion, this type of work is suitable for publication in Polymers but a revision must be performed before publication. Thus, the text must be extensively revised to correct some English grammar (e.g., “they thus plays” (line 29) -> “they thus play”) and several typos (e.g., “10^8 cells/cm, The uniform” -> “10^8 cells/cm. The uniform” (line 27); or “deformation. the stress” -> “deformation. The stress” (line 260); or “which is easy to cause easy to cause” (line 262)). Also, in some figures (e.g., figures 11 and 13) Chinese symbols should be changed to Latin symbols. On the other hand, authors could explain the meaning of eq. 1 and justify why it is written there. Likewise, authors could mention why they use certain parameter values and not others: why the strain was set 1% and the frequency 0.1 Hz for the rheological test (section 2.3.2)?
Response: Thanks for your good advice. Based on your suggestion, We have made a lot of modifications in the revised manuscript:
Point 1:The text must be extensively revised to correct some English grammar.
Response 1: Sorry for our poor expression. According to your suggestion, we have revised this paragraph to make the sentence clear,it has been marked in blue in the revised manuscript.
Point 2: In some figures (e.g., figures 11 and 13) Chinese symbols should be changed to Latin symbols.
Response 2: Thank you for correcting some our mistakes.Figures 11 and 13 have been modified in the revised manuscript. (see Figures 18 and 20)
Point 3: On the other hand, authors could explain the meaning of eq. 1 and justify why it is written there.
Response 3:Thanks for your good suggestion.we have used language instead of formula in the background of the revised manuscript.
Point 4: Authors could mention why they use certain parameter values and not others: why the strain was set 1% and the frequency 0.1 Hz for the rheological test (section 2.3.2)
Response 4:Thanks for your good advice.The strain was 1% to ensure that polypropylene was in the linear viscoelastic region.The frequency of 0.1Hz is mainly to ensure the stability of the material in the process of detection.

Reviewer 3 Report
The authors studied the foam bubble deformation during the injection molding process and analyzed the geometry of the bubble using visual measurements and build a computational analysis of temperature effect on two neighboring bubbles in the PP matrix during the temperature change. They found that the deformation of isolated cells depends on temperature but no macroscopic deformation. The temperature effect also caused internal stress and cause some shape changes to the cells. The purpose of the study seems to compare the experimental findings and computational results and try to understand how the bubbles deform upon temperature change. However, the logic of the manuscript does not match very well between the experimental facts and the FEM computational model. Not only that, but the conclusion of the work is also very confusing and didn’t put many efforts onto it. I don’t suggest this paper published on Polymers unless the authors address the following issues:
(1) The visual observation on one isolated cell shows there is no variation in the cell deformation upon temperature change. However, the computational model is focusing on two neighboring cells. Why setup two different systems and compare between them. It looks like the work is unfinished and need more data to draw any conclusion. It is also logical meaningless to compare two different systems using two different methods. I just feel confused when I finish the reading and the conclusion does not make sense to me. To make it logically sound, one must compare the same system using two different methods. If the two methods give the same results, then one can drawn some conclusions. Please adding more data to make it meaningful before drawing any conclusions.
(2) If I understand correctly, the polymer melt was injected into the mold at different temperatures and the camera was set in the mold and capturing the images of the bubble growing. My confusion is it looks like there is no heating element at the molding region and the whole sample is undergoing a cooling process (mold temperature is only 40 C) while the foaming reaction happened. Does it really make sense to study an isolated cell growing during the cooling process? Considering the results (no visible change in the deformation degree D among the three temperature), why the authors chose to study the dependence of deformation degree D upon the injection temperature. The initial temperature does affect the cooling dynamics, but it is seems like the cooling process is more important affecting the cell growth.
(3) Associated with (2), the computational model is built based on the injection molding temperature where the foaming process just started. It is not the same condition for visualizing the cell deformation as well. Please redesign your system and try to find the same target to study. It just makes no sense to design a project like this.
(4) In fig 12, the authors provided a plot about the local stress value vs. length. Could the authors clarify the definition of the length? It is hard to imagine how the stress values were sampled. Please label them on fig 11 as well. In fig 12, I also noticed a jump on the curve at T=205 C. Is there anything wrong with the data? Why the data points are missing. Similar issue also happened in fig 14. Please check them as well.
(5) The language needs a deep polishing, so many typos and confusing sentences are found.
Author Response
We must thank you and the reviewer for the critical feedback on our manuscript entitled "Deformation and Simulation of the Cellular Structure of Foamed Polypropylene Composites" (Manuscript Number:polymers-1984982). We really appreciate all the valuable comments from the reviewers, which not only helped us to improve the quality of our manuscript, but also suggested some good ideas for future studies. Now we have added some experimental results and carefully revised the whole manuscript according to reviewers’ comments. The revised manuscript has been highlighted in blue in the revised manuscript. Below you will find our point-to-point responses to the reviewers’ comments and questions.
Response to Reviewer 3 Comments
Point 1: The visual observation on one isolated cell shows there is no variation in the cell deformation upon temperature change. However, the computational model is focusing on two neighboring cells. Why setup two different systems and compare between them. It looks like the work is unfinished and need more data to draw any conclusion. It is also logical meaningless to compare two different systems using two different methods. I just feel confused when I finish the reading and the conclusion does not make sense to me. To make it logically sound, one must compare the same system using two different methods. If the two methods give the same results, then one can drawn some conclusions. Please adding more data to make it meaningful before drawing any conclusions.
Response 1: Thanks for your good suggestion.The main purpose of the deformation analysis of isolated cell is to provide a basis for the deformation analysis of adjacent cells. Through the analysis of isolated cell, it is found that the cell deformation parameter (D) ≤ 0.018, which is the basic value for analyzing the deformation of adjacent cells. The deformation law of adjacent cells obtained in this way has practical value.At the same time,We have used two different methods to compare the same system ,the matching experimental results and analysis have been supplemented in manuscript 3.2.
Point 2: If I understand correctly, the polymer melt was injected into the mold at different temperatures and the camera was set in the mold and capturing the images of the bubble growing. My confusion is it looks like there is no heating element at the molding region and the whole sample is undergoing a cooling process (mold temperature is only 40 C) while the foaming reaction happened. Does it really make sense to study an isolated cell growing during the cooling process? Considering the results (no visible change in the deformation degree D among the three temperature), why the authors chose to study the dependence of deformation degree D upon the injection temperature. The initial temperature does affect the cooling dynamics, but it is seems like the cooling process is more important affecting the cell growth.
Response 2: Thanks for your good advice.The foaming process was a kind of instantaneous supercritical pressure release process,the mold temperature is really important to the final shape of the cell. However, the foaming process of the mold cavity is completed instantaneously, and the molding time is about 0.5s, which has little influence on the cell deformation , the mold temperature mainly has a significant influence on the surface quality of the foaming products.Therefore, we set the mold temperature to 40℃in this work, the three temperatures are mainly considered to affect the melt viscosity of PP materials, we didn't consider the heating element for the molding region.
Point 3: Associated with (2), the computational model is built based on the injection molding temperature where the foaming process just started. It is not the same condition for visualizing the cell deformation as well. Please redesign your system and try to find the same target to study. It just makes no sense to design a project like this.
Response 3: Thanks for your good suggestion.Our calculation model is based on injection molding process conditions and material properties,the initial radius of 100μm is set according to the average radius of the visualized cell,The internal pressure of 100Pa in the cell is calculated according to the ideal gas equation ,the shear modulus and viscosity of the model are set according to the rheological properties of the material at different temperatures,the whole simulation is basically consistent with the actual injection conditions.
Point 4:In fig 12, the authors provided a plot about the local stress value vs. length. Could the authors clarify the definition of the length? It is hard to imagine how the stress values were sampled. Please label them on fig 11 as well. In fig 12, I also noticed a jump on the curve at T=205 C. Is there anything wrong with the data? Why the data points are missing. Similar issue also happened in fig 14. Please check them as well.
Response 4:Thanks for your good suggestion."Length" is the circumference of the cell model, and the length is 628μm, which we have indicated in figure 21 and use language to explain.The data points in Figures 11 and 12 are wrong, and we have modified them in Figures 18 and 20 .
Point 5:The language needs a deep polishing, so many typos and confusing sentences are found.
Response 5:Sorry for our poor expression. According to your suggestion, we have revised this paragraph to make the sentence clear,it has been marked in blue in the revised manuscript.

Reviewer 4 Report
1. The entire abstract section must be revised to give a brief explanation of the importance, investigations, and outcomes with the advantages/significance of this research study. Also, the novelty of the study should be reflected in the abstract.
2. The authors need to highlight the novelty of the work presented. Please, demonstrate more obvious your innovation in this work. Would you mind identifying blatant discrimination between your work and others?
3. I also suggest improving the literature review by discussing those mentioned above and other journal papers facing similar issues. the followings are some of these references:
https://doi.org/10.3390/polym11060953
https://doi.org/10.3390/polym11010100
https://doi.org/10.1080/15376494.2021.1972496
4. The introduction section is not up to the mark. In the introduction section, you need to connect the state of the art to your paper goals only. Hence modify the entire section accordingly and present the specific goals/research objectives in the last part of the introduction section.
5. A flowchart should be provided for the work process.”
6. please add witness lab photos.
7. No Error % was found for the models.
8. Conclusions, not conclusion?.
9. Conclusions have to be numbered?
Author Response
We must thank you and the reviewer for the critical feedback on our manuscript entitled "Deformation and Simulation of the Cellular Structure of Foamed Polypropylene Composites" (Manuscript Number:polymers-1984982). We really appreciate all the valuable comments from the reviewers, which not only helped us to improve the quality of our manuscript, but also suggested some good ideas for future studies. Now we have added some experimental results and carefully revised the whole manuscript according to reviewers’ comments. The revised manuscript has been highlighted in blue in the revised manuscript. Below you will find our point-to-point responses to the reviewers’ comments and questions.
Response to Reviewer 4 Comments
Point 1:The entire abstract section must be revised to give a brief explanation of the importance, investigations, and outcomes with the advantages/significance of this research study. Also, the lty of the study should be reflected in the abstract.nove1.
Response 1:Thanks for your good suggestion.We have added the strengths/implications of this study in the abstract,it has been marked in blue in the revised manuscript.
Point 2: The authors need to highlight the novelty of the work presented. Please, demonstrate more obvious your innovation in this work. Would you mind identifying blatant discrimination between your work and others?
Response 2: Thanks for your good suggestion.We have added novelty to the manuscript, see lines 28-30.
Point 3: I also suggest improving the literature review by discussing those mentioned above and other journal papers facing similar issues. the followings are some of these references:
https://doi.org/10.3390/polym11060953
https://doi.org/10.3390/polym11010100
https://doi.org/10.1080/15376494.2021.1972496
Response 3: Thanks for your good advice.Among the three journal papers provided by the reviewers, two journal papers mainly study the mechanical behavior of foaming materials, and one has been added in the introduction, see lines 85-91.
Point 4: The introduction section is not up to the mark. In the introduction section, you need to connect the state of the art to your paper goals only. Hence modify the entire section accordingly and present the specific goals/research objectives in the last part of the introduction section.
Response 4: Thanks for your good advice.We have added some content in the introduction section.
Point 5: A flowchart should be provided for the work process.
Response 5: Thanks for your good suggestion.We've added the workflow flowchart of the manuscript in "highlight".
Point 6: please add witness lab photos.
Response 6: Thanks for your good suggestion.The lab photos have been supplemented in manuscript 3.2.
Point 7: No Error % was found for the models.
Response 7: Thanks to the review experts for our approval
Point 8: Conclusions, not conclusion?
Response 8: Thanks for your good suggestion.We have made some modifications to the conclusion
Point 9: Conclusions have to be numbered?
Response 9: Thanks for your good suggestion.The conclusion is not numbered.

Round 2
Reviewer 1 Report
The authors have addressed satisfactorily the points raised during the review
Author Response
We gratefully appreciate for your comments. We have revised the introduction of the manuscript, and added several relevant references. Such as page 2, lines 31, 34; page 3, lines 81 to 86, 97. According to the Reviewer’s suggestion, we have re-written the conclusion based on the results in the page 16, lines 311 to 319. All the changes marked in blue in the paper, which we hope meet with approval.
Reviewer 3 Report
Please put more effort on polishing the figures in the manuscript. The plotting style (size, resolution, line color, line style etc) varied quite a lot for different figures. Fig 21 still has the jump at the end of the points. The line and should match the symbol. If the displacement is measured along the circumstance of the circle, why they have different value at two ends, if I understand correctly, the curve should be able to connected at the ends since it is a continuous measurement.
Author Response
Thanks for your good suggestion.Since the graduate has graduated, some of the drawings in the manuscript can't find the original data for drawing, and the drawing style (size, resolution, line color, line style, etc.) can 't be modified in different graphics, so I hope to bring the reviewer's understanding.According to the suggestion of reviewer, we have modified Fig 21. Because there is a certain error in the displacement between 458μm-628μm, Fig 21 shows only the displacement variation of adjacent cells between 0-458μm, so the curve can't connected at the ends. For the language of the article, we have already found someone to polish it.
Reviewer 4 Report
Accept in present form.
Author Response
We would like to thank you for your careful reading, helpful comments, and constructive suggestions, which has significantly improved the presentation of our manuscript. We have checked the spelling and polished the English language. All the changes marked in blue in the paper, which we hope meet with approval.